# GraphPCB: Graph-encoded Printed Circuit Board Datasets for Component Classification with Graph Neural Networks

## Abstract

We present a graph-based framework for Printed Circuit Board (PCB) image analysis, targeting core hardware assurance tasks such as IC segmentation and component identification. PCB images differ fundamentally from natural images in texture simplicity, spatial sparsity, and lack of informative backgrounds, limiting the effectiveness of traditional vision models. We propose GraphPCB, a generic scheme that transforms PCB images into graph-structured data, where nodes represent localized component regions and edges encode spatial proximity. This representation enables the application of Graph Neural Networks (GNNs) to PCB understanding, offering robustness to geometric variations and background noise. We release two high-quality GraphPCB datasets and analyze their structural properties, including graph heterophily and domain-specific challenges. Extensive experiments with various GNN architectures provide benchmarks and insights, establishing GraphPCB as a new testbed for node classification in structured visual domains.

## 1 Introduction

Printed Circuit Boards (PCBs) constitute the structural and functional backbone of contemporary electronic systems and are deployed extensively across diverse industrial domains. The operational performance and long-term reliability of these systems are intrinsically linked to PCB quality, necessitating rigorous inspection protocols to ensure hardware correctness and system robustness (Kim et al., 2005; Akhyar et al., 2022; Cheng et al., 2023; Chen et al., 2024; Ou et al., 2024). A PCB board usually contains many different (functional) types of electronic parts. Integrated Circuits (ICs), which perform essential computational and signal processing tasks, are among the most critical components on PCBs and demand precise identification during inspection. PCB image processing tasks, such as IC segmentation and electronic component identification (Makwana et al., 2023; Wang et al., 2025), are integral to Hardware Assurance (HA) frameworks. They play a pivotal role in verifying design integrity, ensuring component authenticity, and maintaining overall system trustworthiness throughout the hardware lifecycle.

PCB images have unique characteristics compared to natural images (see Figure 1 for an example). The electronic components usually have much simpler visual features, such as more regular shapes and simpler textures. Therefore, their image features are less informative. Moreover, position-wise, each component is usually placed in isolation from other components. This is because they are interconnected internally, and the connecting metal lines are usually hidden. Therefore, the relative isolation of the components makes it more difficult to fathom the semantic meaning of the image as a whole. Fundamentally, PCB images have a (design) logic for the layout of their foreground objects that is distinct from natural images. In view of the above characteristics, we propose using graph-based models (Shuman et al., 2013; M. Defferrard, 2016; Kipf & Welling, 2017; Veličković et al., 2018), in addition to conventional image processing tools (He et al., 2016), for analyzing PCB images.

The *graph-based approach* has several advantages. First of all, the analysis of PCB components should be independent of the orientation and scaling of the image. Such properties of invariance under geometric transformations are well-observed for graph-structured data. Moreover, unlike

natural images, the background of a PCB image usually does not provide much useful semantic information. Differences in lighting, color, and background contrast on different boards may even confuse a computer vision (CV) model during training. On the other hand, a graph-structured data retains only visual information of the PCB components and their relative positional information. The potentially harmful distribution shift of background visual information can thus be avoided.

It is observed in Wang et al. (2025) that the critical step in IC segmentation is to correctly distinguish IC components from visually similar components such as discrete transistors. With the graph-based setting, the analysis of PCB components can thus be interpreted as the *node classification problem* for graph neural networks (GNNs). Novel high-quality datasets are needed to investigate whether using graphs is beneficial for PCB image processing. Therefore, the main objective of our paper is to propose a general scheme that converts PCB image datasets into graph datasets, generating graph-encoded PCB datasets *GraphPCB*.

For a high-level overview, we partition an image into "cells" of PCB components, and each cell is associated with a node. Nodes of adjacent cells are connected by an edge in the graph. The construction preserves the relative positional information of the components. Encoded image features of each cell are used as node features. Details shall be provided in Section 3.

Our main contribution can be summarized as follows:

- Conceptually, we give a new perspective for PCB image processing based on graph learning. We release two new GraphPCB datasets based on the proposed approach.

- For the GNN community (M. Defferrard, 2016; Kipf & Welling, 2017; Veličković et al., 2018), we thoroughly analyze the GraphPCB datasets, including properties such as graph heterophily, which are of great interest to the community. We highlight their differences with commonly used node classification datasets. In terms of both application domain and dataset characterizations, GraphPCB datasets contribute in a new way to the GNN data repository.

- We evaluate the performance of different GNN models on the datasets and discuss their pros and cons. The results can be used as baselines for future research on these datasets.

- The generic scheme of converting an image into a graph proposed in the paper may find applications outside the subfield of PCB image processing.

The datasets and source code are provided in the supplementary materials.

## 2 RELATED WORKS

PCB image processing has received considerable attention in recent years. For example, Li et al. (2013) considers the segmentation of surface-mounted devices on PCBs for automated recycling. Cheng et al. (2024) considers the automatic detection of such devices utilising YOLO object detection models. Huang et al. (2022) uses a semantic segmentation method to extract high-level semantic features of PCB assembly images for applications such as component detection. For IC segmentation, in the recent work Wang et al. (2025), it is observed that a major challenge for IC segmentation is to decide visually whether a given component is IC. This is a difficult task even for human experts in the domain of HA. This inspires us to consider a graph-based approach, so that one may leverage information of neighboring components and uncover the implicit design logic of PCB boards. Therefore, to facilitate research in this direction, it is imperative to create and release high-quality datasets.

On the other hand, node classification is a common task for GNN (Kipf & Welling, 2017). Most early public datasets are on citation networks (Sen et al., 2008). These datasets exhibit strong *homophily* (Zhu et al., 2021; Wang et al., 2022), i.e., edges are likely to connect nodes of the same type. *Heterophilic datasets* without such a property are subsequently released (Pei et al., 2020; Platonov et al., 2023), and they include networks for citation, co-purchasing, web link, and social relations. New GNN models dedicated to such datasets have been proposed (Platonov et al., 2023; Luan et al., 2022; Rusch et al., 2022; Kang et al., 2024; Lee et al., 2024), and this research area remains active. There are also datasets that are more application-oriented. For example, Kearnes et al. (2016); Xu et al. (2019) has introduced graph datasets for biochemistry; however, the major

task is graph classification or regression. In HA, there are (node classification) datasets and research works for netlist analysis (Hong et al., 2024; Zhang et al., 2025). As far as we are aware, ours is the first work that releases and thoroughly analyzes *graph datasets* for PCB images. We hope that it can benefit both the HA and GNN communities.

## 3 THE GRAPHPCB DATASETS

In this section, we describe a general scheme that converts a PCB image dataset into a graph dataset, which includes graph construction and the generation of node features and labels. We apply the scheme to construct two graph datasets for node classification. We further analyze their properties, focusing on those of interest to the GNN community.

### 3.1 GRAPH CONSTRUCTION

In this subsection, we describe how to convert a PCB image $\mathcal{I}$ into a graph $G = (V, E)$. As an overview, our graph construction strategy, which encodes local spatial connections, is inspired by layout principles that are closely aligned with DFM rules. For example, Rothstein's "*20 Guidelines for Efficient PCB Component Placement*" emphasizes grouping functionally related components to reduce trace lengths and enhance signal integrity—principles that often manifest as spatial proximity in real-world PCB layouts. Other graph construction methods will be explored in Appendix D.

In essence, we want to *faithfully* represent the floorplan of the PCB design to classify key components, and hence, the relative positions of the components should be preserved as much as possible. For this, we propose the following construction based on *Voronoi tessellation* (Aurenhammer, 1991) (see Figure 1 for an example):

- Each node $v \in V$ corresponds to an electronic component on the image $\mathcal{I}$, and we denote its *center* by $x_v$.
- We partition the image into regions called *Voronoi cells* $R_v$, each associated with a node $v \in V$. Specifically, for each point $x$ on the image, it belongs to the region $R_v$ if the following holds:
$$\|x_v - x\| \leq \|x_{v'} - x\|,$$
for any $v \neq v' \in V$. Intuitively, $x$ is assigned to its *closest* component(s). As a result, the image is partitioned into convex cells $R_v, v \in V$.
- We form an edge $(v, v') \in E$ between a pair of nodes $v, v'$ if their cells $R_v$ and $R_{v'}$ are adjacent to each other, i.e., $R_v$ and $R_{v'}$ share a common boundary.

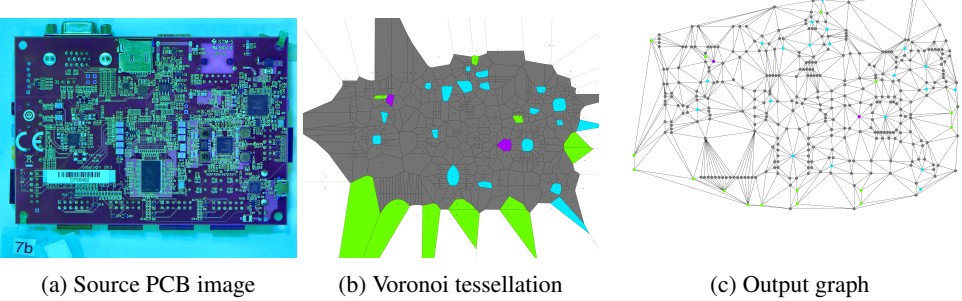

| (a) Source PCB image | (b) Voronoi tessellation | (c) Output graph |

Figure 1: This is an illustration of the graph generation procedure. A source PCB image (in (a)) is converted into a tessellation of Voronoi cells (in (b)). Each cell corresponds to a PCB component and hence a node in the output graph (in (c)). An edge connects a pair of nodes if their associated cells share a common boundary line. For the color scheme, "cyan", "purple", "green" and "grey" are for the classes "IC", "DT", "Diode" and "others" respectively.

The procedure converts the image $\mathcal{I}$ to a *planar* graph $G$. This means that $G$ can be embedded in the plane. Moreover, if two components are close to each other on the image, their corresponding

nodes in $G$ tend to be connected by an edge as their cells are likely to share a common boundary. Therefore, the construction preserves the relative locations of the components while discarding less useful background information.

## 3.2 FEATURES AND LABELS

For node classification, nodewise labels and features/attributes are essential for training and inference. We describe how they are curated in this subsection.

Recall that each node $v \in V$ corresponds to an electronic component in the image $\mathcal{I}$. Therefore, it is natural to use the image features of the component as the node feature of $v$. Specifically, following Wang et al. (2025), the features are extracted using a pre-trained ResNet50 encoder (Deng et al., 2009; He et al., 2016). The model is sufficient for visual feature extraction due to the simplicity of PCB components.

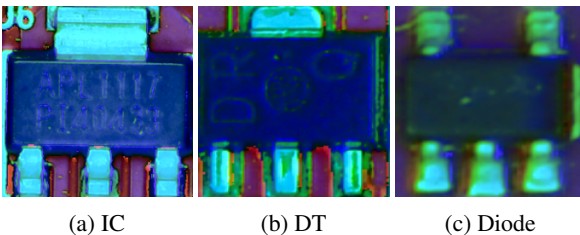

(a) IC       (b) DT       (c) Diode

Figure 2: Sample images of IC, Discrete transistor (DT), and Diode component, which can be visually very similar. This is the main challenge for identifying IC components.

For each PCB dataset, there is a long list of types of electronic components (see Appendix A). However, not all component types are of equal importance for hardware assurance. In particular, identifying integrated circuit (IC) components is of paramount importance (cf. Section 1). This is usually a challenging task, as there are other component types that are visually similar to ICs. Our primary objective is to:

- *Distinguish ICs from discrete transistors (DTs) and diodes, which can be visually similar (cf. Figure 2).*

Therefore, we propose to consider the *label set* $\mathcal{C}$ consisting of 4 classes: *"IC", "DT", "Diode", and "others"*. As the "others" class consists of many different types of components, it is very large in size. Moreover, the features of nodes belonging to this class are highly diversified, which may pose additional challenges.

The procedure described above is applied to two publicly available PCB image datasets: FPIC (Jessurun et al., 2023) and WACV (Kuo et al., 2019). The resulting graph datasets are named *Graph-F* and *Graph-W* respectively. Notice that, unlike most well-studied GNN datasets for node classification, each dataset contains multiple graphs, each of which is constructed using one image from the image dataset.

For data split, we follow approximately the same training/testing ratio $0.7/0.3$ as in Jessurun et al. (2023). Details are given in Appendix A.

## 3.3 DATASET PROPERTIES

In this subsection, we analyze structural properties of the Graph-F and Graph-W datasets. We focus on two main aspects:

- Geometric properties of the graphs.
- The relations between the graph structure and node labels.

In Figure 3 and Figure 4, we show fundamental graph properties of the datasets, including *average degrees*, *clustering coefficients*, and *diameters*. Recall that, unlike most well-known node classifica-

tion datasets, both Graph-F and Graph-W contain multiple graphs. Therefore, a subfigure contains the distributions of an attribute across all graphs in a dataset.

We notice that both datasets have *similar general trends* for reported graph features, while Graph-F has *more outliers*. For example, for both datasets, the average degree is concentrated near 6, while Graph-F contains a few graphs with very low degrees. The observations suggest that the distributions of the (reported) graph attributes for both datasets are generally aligned, which may be due to both datasets following the same design logic for PCB layout.

For each dataset, the sizes of the graphs are *diverse*. As the classification task is inductive, i.e., training and testing are performed on different graphs, the graph diversity may pose additional challenges.

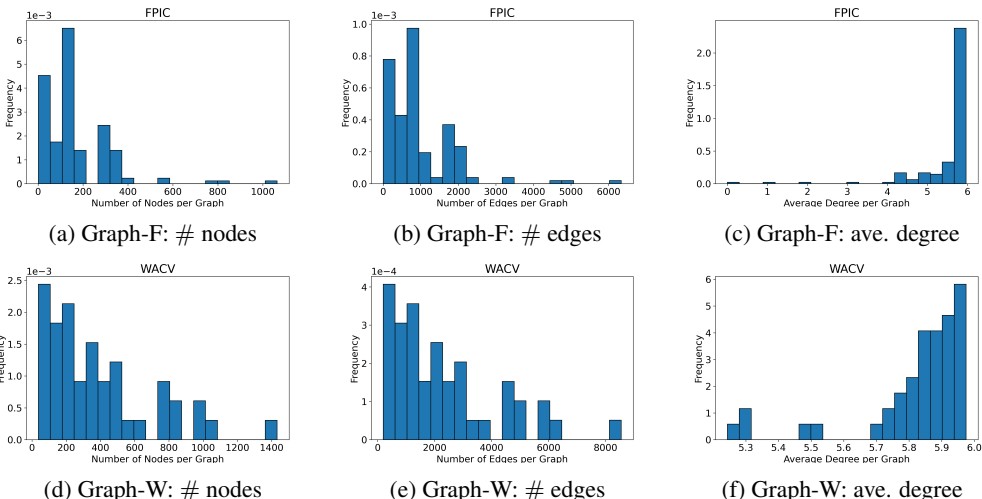

Figure 3: Distribution of basic graph statistics for Graph-F and Graph-W datasets.

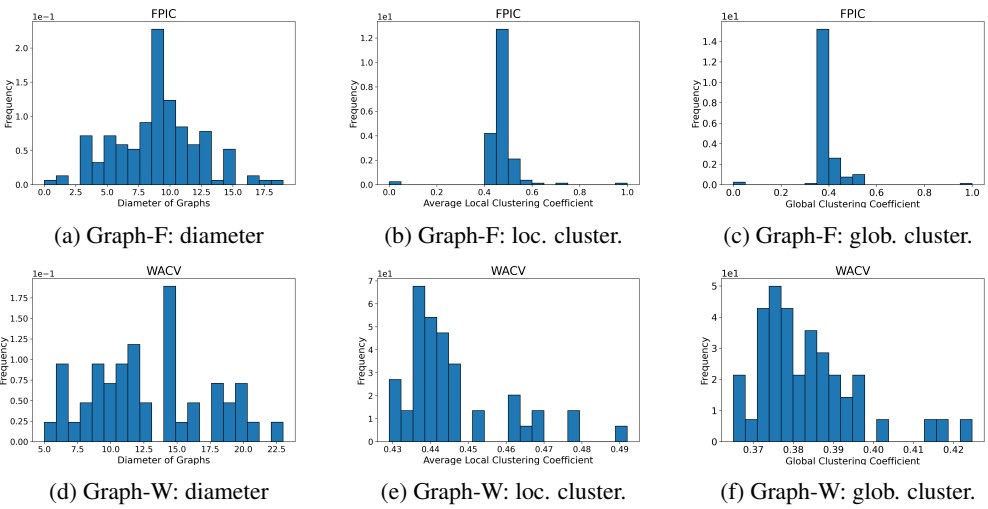

Figure 4: Distribution of graph diameters and local/global clustering coefficients for Graph-F and Graph-W datasets.

For graph-label relations, we report *edge homophily, adjusted edge homophily, and label informativeness (LI)* (introduced in Platonov et al. (2023; 2024) in Figure 5). Edge homophily estimates the percentage of edges connecting nodes with the same label. Intuitively, if a dataset has a low homophily score, then there are many edges connecting pairs of nodes from different classes. One refers to such a dataset being *heterophilic*. Message-passing neural networks, such as GCN, require substantial modification for such a dataset (Platonov et al., 2023). Adjusted edge homophily is a

modification of edge homophily for class-imbalanced graph datasets. On the other hand, LI quantifies how much information a node receives from the labels of its neighbors.

Due to class imbalance (the large "others" class), graphs in both datasets generally have high edge homophily (cf. Figure 5). However, this is rectified by the adjusted homophily score, which is generally small for both datasets. This suggests that intrinsically, the graphs are heterophilic. This is reasonable as PCB components are usually not grouped according to their types. Label informativeness (LI) is generally high for both datasets, which means that for a node, it is viable to leverage information from its neighbors for its own classification. However, there are graphs with very low LI, as there are graphs of small sizes (cf. Figure 3), making it harder to draw useful information from neighbors.

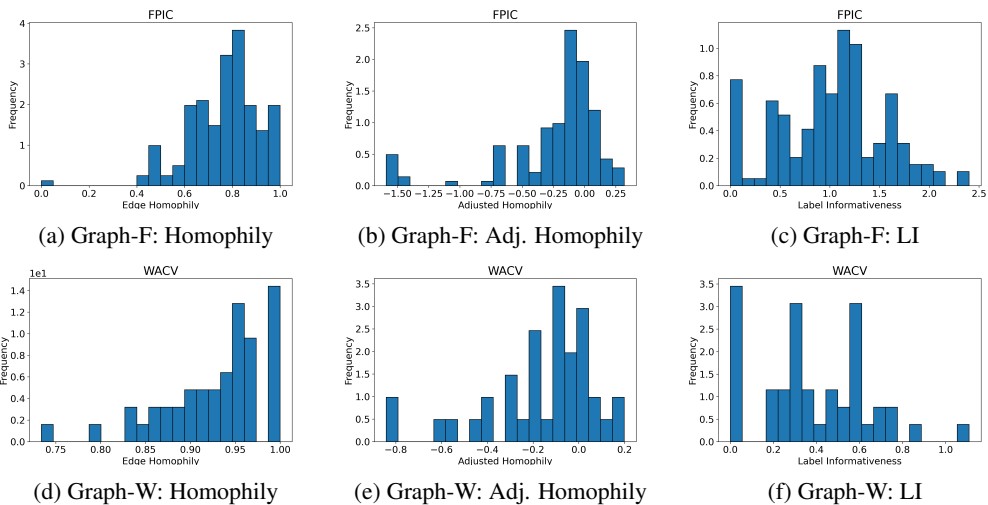

(a) Graph-F: Homophily (b) Graph-F: Adj. Homophily (c) Graph-F: LI

(d) Graph-W: Homophily (e) Graph-W: Adj. Homophily (f) Graph-W: LI

Figure 5: Distribution of (edge) homophily, adjusted (edge) homophily, and LI for Graph-F and Graph-W datasets.

We summarize the observations in Table 1, and compare with other commonly used datasets (a representative is selected for each reference). We notice that compared with any existing dataset, GraphPCB datasets always have some distinctive traits. The table can be used (a) to identify the challenges of node classification on the GraphPCB datasets, and (b) to guide the search for suitable models to tackle the task, which we will do in the next section.

Table 1: A comparison of dataset properties. The observations for GraphPCB are reported based on the average of all graphs in the datasets. Compareing datasets: Cora (Sen et al., 2008), Cornell (Pei et al., 2020), Roman-empire (Platonov et al., 2023), Proteins (Xu et al., 2019).

| Datasets | GraphPCB | Cora | Cornell | Roman-empire | Proteins |
|---|---|---|---|---|---|
| Node classif. | ✓ | ✓ | ✓ | ✓ | ✗ |
| Multi. graphs | ✓ | ✗ | ✗ | ✗ | ✓ |
| Inductive | ✓ | ✗ | ✗ | ✗ | ✓ |
| High diversity | ✓ | —— | —— | —— | ✗ |
| Heterophilic | ✓ | ✗ | ✓ | ✓ | —— |
| High LI | ✓ | ✓ | ✗ | ✗ | —— |

# 4 EXPERIMENTAL RESULTS

In this section, we propose GNN baselines for node classification on the datasets: Graph-F and Graph-W. The performance is analyzed, and the results can be used as benchmarks for future research on these datasets. We show run-time and discuss other graph constructions in Appendix C, D

### 4.1 BASELINES

As we use an image encoder to generate node features, an MLP is equivalent to the CV baseline for the task. In particular, it does not use the graph structure. For graph-based models, we first include standard GNN node classification models, GCN, GAT (Kipf & Welling, 2017; Veličković et al., 2018) as baselines. We also consider GraphSAGE (Hamilton et al., 2017), a popular model for inductive learning. We implement three versions: Softmax (S-M), Standard deviation (Std), and Attentional (Att), according to different aggregation mechanisms.[1]

From Section 3.3, we notice that the graphs are generally heterophilic. Therefore, we include GAT-sep and GT-sep proposed in Platonov et al. (2023). Here, "-sep" refers to the idea of separating ego- and neighbor-embeddings in the GNN aggregation step proposed in Zhu et al. (2020), and GT is the graph transformer model (Dwivedi & Bresson, 2021). We include ACM-GCN (Luan et al., 2022), which is verified to have superior performance for heterophilic datasets in both Lee et al. (2024) and Ji et al. (2025). As explained in Sun et al. (2023), a node classification problem can also be cast as a graph classification problem by viewing a neighborhood of each node as a (sub)graph. This point of view agrees with the insight to leverage neighboring node types for the classification. Therefore, we also include the graph classification baseline GIN (Xu et al., 2019).

To overcome label imbalance, we use the *weighted binary cross entropy (BCE) loss* for training, where IC, DT, and diode carry a higher weight inversely proportional to their occurrence.

### 4.2 EVALUATION

We have mentioned that the node classes of both datasets are highly *imbalanced* (see Appendix A). As a union of many different component types, the "others" class is substantially larger in size. Therefore, we propose to evaluate model performance using the *F1-score*, which is the harmonic mean of the *recall* and *precision*.

As we are mainly interested in the IC components, we want to evaluate whether a model can tackle the confusion among "IC", "DT" and "Diode" classes. To this end, we report the *Subset F1-score*, which excludes the "others" class from the evaluation. Specifically, the Subset F1-Score is computed by slicing the confusion matrix to retain only the rows and columns corresponding to the selected classes, completely ignoring class 3 in both the ground truth and predictions. This metric provides a more targeted and meaningful assessment of the model's performance on the primary classes of interest, ensuring that the evaluation aligns with the specific goals of the task.

In this work, we want to explore whether a graph-based method can capture useful structural information (over visual details) for the classification task. For this, we report the *percentage of overlapping detections (POD)* (a metric used in Latif et al. (2023)) between the CV baseline (MLP) and each GNN baseline, for "IC", "DT", and "Diode" classes. A *smaller* POD indicates more distinct information provided by the graph approach.

### 4.3 RESULTS AND DISCUSSIONS

From the scores reported in Table 2, we observe that direct feature aggregation via message passing (GCN, GAT, or GIN) performs poorly. It is likely due to the heterophilic nature of the datasets. Different versions of GraphSAGE show consistent good performance for both datasets, which suggests GraphSAGE is a good candidate for such a task. It verifies the effectiveness of GraphSAGE in inductive settings. Dedicated models for heterophilic datasets are strong competitors, particularly for the Graph-W dataset. These models show less confusion among the three main classes. The observations generally agree with our findings in Section 3.3 regarding the dataset properties.

For POD shown in Table 3, we notice the general trend that POD for "IC" is relatively higher than "DT" and "Diode". Nevertheless, they show substantial disagreement with MLP, particularly for the "Diode" class. This suggests that the graph-based approaches can extract structural information in addition to visual information, resulting in different predictions. This may enhance our understanding of the patterns in PCB design.

---

[1] `https://pytorch-geometric.readthedocs.io/en/latest/modules/nn.html?highlight=torch_geometric+nn+aggr#aggregation-operators`

Table 2: The F1- and Subset F1-scors for the benchmarks (GPU: A5000, Memory: 24G). **Red** indicates the highest, **blue** the second highest performance in each column.

| Model | Graph-F | | Graph-W | |
| --- | --- | --- | --- | --- |
| | **F1-Score** | **Subset F1-Score** | **F1-Score** | **Subset F1-Score** |
| MLP | 0.68 | 0.77 | 0.49 | 0.61 |
| GCN | 0.44 | 0.72 | 0.36 | 0.53 |
| GAT | 0.44 | 0.76 | 0.17 | 0.52 |
| GIN | 0.41 | 0.60 | 0.31 | **0.69** |
| GraphSAGE (S-M) | **0.71** | 0.81 | 0.54 | 0.60 |
| GraphSAGE (Std) | **0.71** | **0.84** | 0.56 | 0.61 |
| GraphSAGE (Attn) | **0.69** | 0.80 | 0.55 | 0.59 |
| GAT-sep | 0.67 | **0.85** | **0.59** | **0.63** |
| GT-sep | 0.65 | 0.83 | **0.58** | **0.63** |
| ACM-GNN | 0.66 | 0.79 | 0.53 | 0.59 |

Table 3: Classwise percentage of overlap detections (POD) calculated over all graphs in the test set.

| Model | POD for Graph-F | | | POD for Graph-W | | |
| --- | --- | --- | --- | --- | --- | --- |
| | **IC** | **DT** | **Diode** | **IC** | **DT** | **Diode** |
| GCN | 0.86 | 0.39 | 0.21 | 0.43 | 0.15 | 0.00 |
| GAT | 0.86 | 0.49 | 0.39 | 0.76 | 0.67 | 0.37 |
| GIN | 0.77 | 0.45 | 0.20 | 0.57 | 0.00 | 0.14 |
| GraphSAGE (S-M) | 0.89 | 0.59 | 0.27 | 0.69 | 0.43 | 0.32 |
| GrapgSAGE (Std) | 0.91 | 0.65 | 0.20 | 0.75 | 0.48 | 0.39 |
| GraphSAGE (Attn) | 0.79 | 0.52 | 0.21 | 0.69 | 0.43 | 0.32 |
| GAT-sep | 0.85 | 0.73 | 0.28 | 0.62 | 0.49 | 0.37 |
| GT-sep | 0.90 | 0.65 | 0.28 | 0.61 | 0.48 | 0.26 |
| ACM-GNN | 0.59 | 0.41 | 0.08 | 0.37 | 0.13 | 0.01 |

## 4.4 POTENTIAL APPLICATION TO IC SEGMENTATION

Accurate IC segmentation is essential for advanced PCB analysis, enabling functional validation, fault localization, and systematic debugging. It is critical for automated electronic inspection systems. To illustrate the potential application of our GNN-based classifier, we consider IC segmentation as a downstream task.

We use SSRNet (Wang et al., 2025) as the backbone. It consists of a coarse segmentation module and an image classifier (see Appendix B for more details). To incorporate the approach of our paper, we keep the coarse segmentation module intact and replace the image classifier with our GNN-based classifier. Based on the overall performance (see section 4.3), we choose GraphSAGE for the classifier, and call the resulting IC segmentation model *SSR-SAGE*.

We remark that SSR-SAGE is *not a rigorous IC segmentation model*, as a full-fledged model requires a module for *object detection*, which is beyond the scope of this paper. The main purpose of the study is to demonstrate the potential benefit of incorporating a graph-based classifier. For this, we illustrate with an explicit test image in Figure 6. More numerical results are shown in Appendix B.

We notice that generic CV models, such as U-Net (Ronneberger et al., 2015), LinkNet (Chaurasia & Culurciello, 2017), fail to generate a clean binary segmentation image. A possible reason is that PCB images have different characteristics as compared with natural images, on which generic CV models are trained (cf. Section 1). Comparing PCBSegClassNet (Makwana et al., 2023) and SSR-NC (SSRNet without a classifier) with SSRNet, we see that a classifier is crucial to remove false-positive IC components. On the other hand, SSR-SAGE demonstrates its ability to correctly

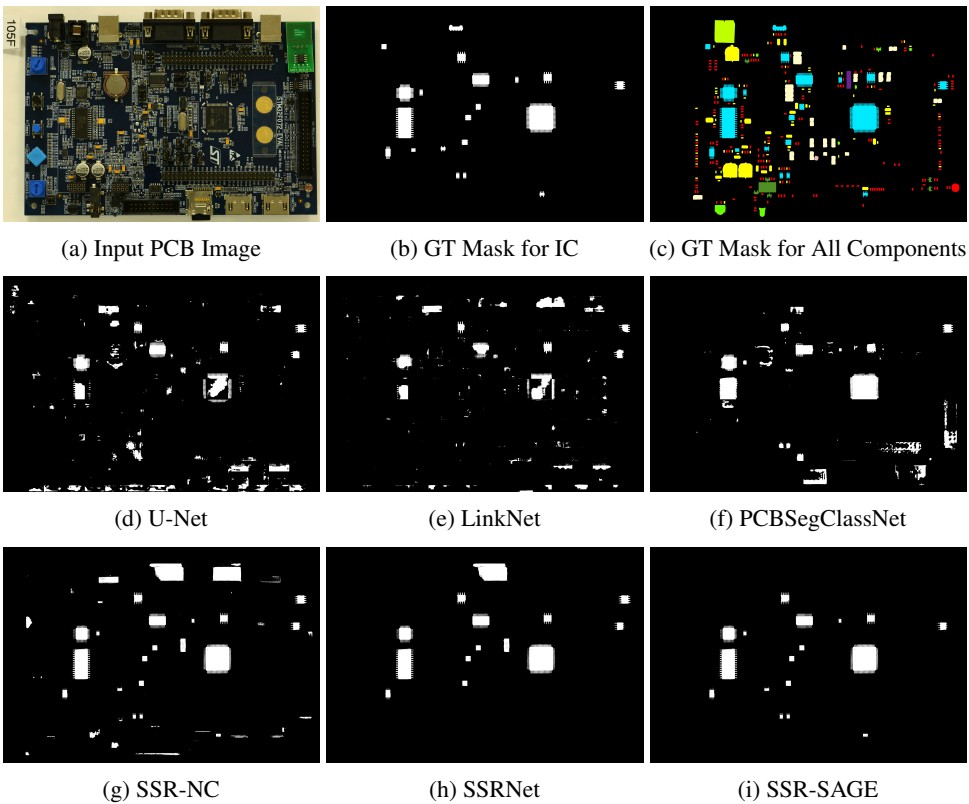

Figure 6: Comparison of different methods for IC segmentation. Row 1 shows the input PCB image and ground truth (GT) masks. (d), (e) are results from generic image segmentation models. (f) is the output of PCBSegClassNet. (g) shows the result for SSRNet without using a classifier, while (i) replaces the image classifier in SSRNet with a graph classifier. SSRNet output is shown in (h).

identify very small IC components as compared with SSRNet. The graph-based classifier indeed offers new insights into the pure image task.

## 5 CONCLUSION

In this paper, we provide a novel perspective on PCB image analysis as node classification in graph-based machine learning. We propose a generic procedure that converts a PCB image dataset into a graph-structured dataset. Two GraphPCB datasets are constructed (and released) and carefully analyzed for their geometric properties and label-edge correlations. Based on the uncovered heterophilic properties of the datasets, a list of GNN benchmarks is applied to the dataset, and their pros and cons are thoroughly discussed based on the performance.

The datasets can be beneficial to both the HA community and the GNN community. On the application side, the classification results can be used for downstream PCB image processing tasks such as IC segmentation, as illustrated in the paper. On the other hand, GraphPCB datasets display unique characteristics as compared with existing node or graph classification datasets, which may facilitate the development of new GNN models.

**Limitations** We have experimented with many existing GNN baselines. However, from the results, we believe there is room for further improvement in terms of the classification performance. To summarize the requirements, a successful model is expected to be able to handle heterophilic graphs in an inductive setting. Moreover, it should be able to aggregate features of different node types in a hybrid manner that takes advantage of both node and graph classification techniques.

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

## A FULL LISTS OF PCB COMPONENTS AND DATA INFORMATION

Table 4: Component types and counts in the FPIC dataset, with GraphPCB labels.

| Component Type | Count | GraphPCB Label |
|---|---|---|
| Resistor (R) | 7246 | 3 |
| Capacitor (C) | 6896 | 3 |
| Integrated Circuit (U) | 761 | 0 |
| Discrete Transistor (Q) | 616 | 1 |
| Connector (J) | 579 | 3 |
| Inductor (L) | 473 | 3 |
| Resistor Coil (RA) | 404 | 3 |
| Diode (D) | 362 | 2 |
| Resistor Network (RN) | 330 | 3 |
| Test Point (TP) | 266 | 3 |
| Integrated Circuit (IC) | 237 | 0 |
| Plug (P) | 200 | 3 |
| Thyristor (CR) | 194 | 3 |
| Motor (M) | 74 | 3 |
| Button (BTN) | 72 | 3 |
| Ferrite Bead (FB) | 69 | 3 |
| CRA | 54 | 3 |
| Switch (SW) | 50 | 3 |
| Transformer (T) | 47 | 3 |
| Fuse (F) | 44 | 3 |
| Vaccum Tube (V) | 41 | 3 |
| Light Emitting Diode (LED) | 39 | 3 |
| Switch (S) | 37 | 3 |
| QA | 36 | 3 |
| Jumper Link (JP) | 31 | 3 |

We show full lists of PCB components and their (total) counts for the FPIC and WACV datasets in Table 4 and Table 5, respectively. Their labels in the converted GraphPCB datasets are given. As many more components belong to the "others" class (with label 3), the datasets are *label imbalanced*.

The datasets introduced in our paper are diverse in terms of PCB types and manufacturing processes, having been sourced from over 38 (Graph-F) or 13 (Graph-W) different companies.

For Graph-F and Graph-W data split, we randomly split the dataset into train/test to approximately 0.7/0.3 ratio (as in Makwana et al. (2023)) for the number of graphs. We check and adjust such

Table 5: Component types and counts in the WACV PCB dataset, with GraphPCB labels.

| Component Type | Count | GraphPCB Label |
|---|---|---|
| Capacitor | 2552 | 3 |
| Resistor | 2271 | 3 |
| Connector | 635 | 3 |
| IC | 400 | 0 |
| Pads | 336 | 3 |
| Pins | 319 | 3 |
| Test Point | 292 | 3 |
| Electrolytic | 251 | 3 |
| LED | 219 | 3 |
| Transistor | 135 | 1 |
| Button | 87 | 3 |
| Jumper | 86 | 3 |
| Diode | 85 | 2 |
| Inductor | 69 | 3 |
| Switch | 60 | 3 |
| EMI Filter | 51 | 3 |
| Relay | 47 | 3 |
| Clock | 37 | 3 |
| Ferrite Bead | 30 | 3 |
| Potentiometer | 9 | 3 |
| Zener Diode | 8 | 2 |
| Fuse | 7 | 3 |
| Display | 6 | 3 |
| Heatsink | 4 | 3 |
| Buzzer | 1 | 3 |
| Battery | 1 | 3 |
| Transformer | 1 | 3 |

that the classes "IC", "DT" and "Diode" are close to 0.7/0.3 ratio in the train/test split. For the IC segmentation study in Section 4.4, we retrain GraphSAGE using the exact same (image) data split as in Makwana et al. (2023).

## B  MORE ON IC SEGMENTATION

We first provide more details on the backbone model SSRNet (Wang et al., 2025). It consists of two key modules: (a) a *few short self-support prototype* segmentation (Fan et al., 2022), and (b) a *region classifier*. The former generates a coarse segmentation with many false positives, i.e., non-IC regions that are visually similar to an IC, while the region classifier uses an *image classifier* to remove the false positive regions. As proposed in Section 4.4, SSR-SAGE replaces the image classifier with the GraphSAGE classifier, while keeping the coarse segmentation (module (a)) intact.

For reference, we show the IC segmentation evaluation metrics *IoU*, *Dice*, *Dice loss*, and *(pixel-wise) error rate* in Table 6. As we have mentioned in Section 4.4, the comparison does not lead to a rigorous conclusion regarding the superiority of SSR-SAGE. Nevertheless, we may observe useful trends. For example, comparing SSRNet with other benchmarks, we see the advantage of using a classifier, which is inherited by SSR-SAGE. The difference in results of SSR-SAGE may be attributed to the GNN classifier that leverages structural information among the PCB components. Therefore, it is worthwhile to consider incorporating graph methods in PCB image processing.

Table 6: A comparison of IC segmentation models.

| Model | IoU ↑ | Dice ↑ | Dice loss ↓ | Error rate ↓ |
|---|---|---|---|---|
| LinkNet | 0.5518 | 0.6734 | 0.3266 | 0.0363 |
| U-Net | 0.5481 | 0.6717 | 0.3283 | 0.0435 |
| DeepLabv3 Chen et al. (2017) | 0.5364 | 0.6658 | 0.3342 | 0.0412 |
| PCBSegClassNet | 0.4246 | 0.5452 | 0.4548 | 0.0434 |
| SSRNet-NC | 0.5759 | 0.6978 | 0.3022 | 0.0345 |
| SSRNet | 0.5957 | 0.7123 | 0.2877 | 0.0319 |
| SSR-SAGE | 0.6619 | 0.7599 | 0.2401 | 0.0224 |

## C  HARDWARE CONFIGURATION AND COMPUTATION OVERHEAD

The hardware setup is: CPU: AMD Ryzen Threadripper PRO 5975WX, 32 Cores; GPU: NVIDIA RTX A5000, CUDA: v12.3, 24GB memory. The configuration is used for obtaining the run-time reported below.

There are 2 main components requiring additional computation: graph construction and GNN. The graph construction is used to produce the datasets. It is not part of any GNN model and can be reused for various downstream tasks. Graph construction is not time-consuming: average 1.49s to convert an image to a graph for FPIC, and 0.81s for WACV.

For GNN, the models we have benchmarked are not complex, and the graphs are generally small. For reference, we show the run-time (in seconds) for the entire training (200 epochs) as:

|  | Graph-F | Graph-W |
|---|---|---|
| MLP | 22.04 | 9.70 |
| GCN | 29.22 | 12.34 |
| GraphSage | 50.86 | 26.23 |
| GT-sep | 103.13 | 32.54 |

None of these models incurs a serious computation burden.

## D  MORE ON THE GRAPH CONSTRUCTION

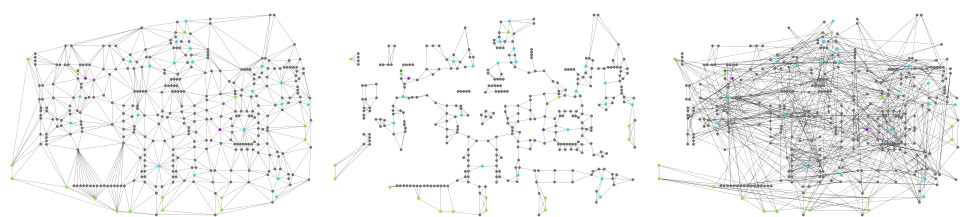

Figure 7: We show the resulting graph from our proposed construction (left), the $k$-NN construction (middle), and the similarity-based construction (right). The $k$-NN construction is similar in spirit to our proposed construction, while our approach more faithfully represents the floor plan of the PCB design. On the other hand, the similarity-based construction may connect components far away from each other.

In this section, we consider other popular graph construction methods: (1) the $k$-NN method (Eppstein et al., 1997) and (2) feature similarity-based graph constructions (Kramer et al., 2008). Briefly, the $k$-NN construction connects each component to its closest $k$-nearest components, measured by the Euclidean distance. It is *similar in spirit* to our approach but less principled. For example, if resistors surround an IC $I$, $k$-NN may exclude some of them from the graph neighbors of $I$. An inherent

limitation of $k$-NN lies in the inhomogeneity of neighborhood structures across components. On the other hand, the similarity-based construction connects pairs of nodes with a large similarity score, usually measured by the cosine similarity of feature vectors. The construction may connect far-away components with similar features. However, this approach may face the same challenge as CV-based approaches, since the construction may connect different and visually similar components.

Table 7: knn: $k$-NN graph construction, sim: similarity-based construction. Best: boldface, 2nd: underscore

| Model | Graph-F | | Graph-W | |
|---|---|---|---|---|
| | F1 | Sub. F1 | F1 | Sub. F1 |
| GraphSAGE-knn | 0.66 | 0.75 | 0.57 | **0.65** |
| GraphSAGE-sim | 0.63 | 0.76 | 0.54 | 0.59 |
| GraphSAGE | **0.71** | **0.84** | 0.56 | 0.61 |
| GT-sep-knn | 0.57 | 0.64 | 0.56 | 0.58 |
| GT-sep-sim | 0.58 | 0.69 | 0.56 | 0.59 |
| GT-sep | 0.65 | 0.83 | **0.58** | 0.63 |

The results are as shown in Table 7. As discussed above, the similarity-based construction often yields suboptimal performance, and feature aggregation via message passing on the similarity graph fails to introduce new information that helps distinguish visually similar components. Compared with the $k$-NN construction, our construction generally leads to better performance for Graph-F. For Graph-W, our approach is comparable to $k$-NN, possibly due to the similar functionality of the graph neighborhoods.

## E  LLM USAGE

We acknowledge the use of large language models (LLMs) as a general-purpose assistive tool in preparing this manuscript. Specifically, LLMs were employed to aid in polishing the writing, including refining grammar, improving clarity, and enhancing fluency of expression. LLMs were **NOT** used for generating research ideas, conducting analysis, or producing results. All conceptual contributions, theoretical developments, experimental designs, and interpretations presented in this work are entirely the responsibility of the authors.

