# OpenReview forum: "GraphPCB: Graph-encoded Printed Circuit Board Datasets for Component Classification with Graph Neural Networks"
_ICLR.cc/2026/Conference — ICLR 2026 Conference Withdrawn Submission_

### Official Review · Reviewer_vWLs · 2025-10-31

**Soundness:** 1
**Presentation:** 3
**Contribution:** 2
**Rating:** 2
**Confidence:** 3

**Summary:**

This paper introduces GraphPCB, a framework that recasts the problem of electronic component classification on PCBs as a graph-based node classification task. The authors argue that traditional computer vision approaches are limited by the unique characteristics of PCB images. Their proposed method converts each PCB image into a graph where nodes represent components and edges represent spatial proximity, determined via Voronoi tessellation. Node features are extracted using a pre-trained ResNet50. The paper's main contributions are: (1) this new graph-based conceptual framework for PCB analysis; (2) the release of two new benchmark datasets, Graph-F and Graph-W, created from public PCB image data; (3) a thorough analysis of these datasets, identifying them as inductive, multi-graph, and heterophilic; and (4) benchmarking of various GNN architectures, demonstrating that models designed for heterophily and inductive learning often outperform baselines. Finally, the utility of the framework is demonstrated by integrating a trained GNN into an IC segmentation pipeline, showing improved performance.

**Strengths:**

- A major contribution is the creation and public release of two new benchmark datasets (Graph-F and Graph-W). The authors provide a thorough statistical analysis of these datasets (graph properties, homophily, label informativeness), which is valuable for the GNN community and provides a solid foundation for future research in this application domain.
- The experimental evaluation is extensive. The authors benchmark a wide range of GNN models, including standard, inductive, and heterophily-aware architectures. This provides a strong set of baselines and offers insights into which model families are suitable for this new task.

**Weaknesses:**

- Marginal performance gains and unexplained results: The significance of the results is questionable on one of the datasets. On Graph-W (Table 2), the best-performing heterophilic GNNs (GAT-sep, GT-sep) achieve a Subset F1-Score of 0.63, which is a very marginal improvement over the simple MLP baseline's 0.61. This minimal gain does not make a strong case for the added complexity of the graph-based approach. Furthermore, the paper fails to discuss the anomalous result where GIN, a simple homophilic model, achieves a Subset F1-Score of 0.69, outperforming nearly all other models on this dataset, which contradicts the heterophily narrative.
- Weak justification for heterophily: The claim that the datasets are intrinsically heterophilic is based on the adjusted homophily score. This score corrects for class imbalance, which is dominated by the large others class. The low score could be an artifact of this single, diverse class connecting to all others, rather than a property of the connections between the key classes of interest (IC, DT, Diode). A more rigorous analysis, such as calculating homophily on a subgraph induced by only these key classes, is needed to substantiate this important claim.
- Ablation study for ResNet50: The paper relies on a fixed, pre-trained ResNet50 for node feature extraction. It is highly plausible that the fixed encoder provides sub-optimal features for this specialized domain. Consequently, the observed performance gains of GNNs over the MLP baseline might not stem from leveraging structural information, but rather from the GNNs' superior ability to process and aggregate these noisy, non-domain-specific features. An ablation study with a fine-tuned encoder is necessary to de-confound these effects and validate the paper's central claim.

**Questions:**

- Could you provide an ablation study comparing results with a fixed ResNet50 encoder versus a fine-tuned one? This would help determine if the GNNs' advantage comes from structural reasoning or from compensating for sub-optimal visual features.
- To strengthen the claim of intrinsic heterophily, can you provide an analysis of homophily (both naive and adjusted) on the subgraph induced by only the key classes of interest (IC, DT, Diode), excluding the others class?

---

### Official Review · Reviewer_nZFC · 2025-11-01

**Soundness:** 2
**Presentation:** 2
**Contribution:** 2
**Rating:** 2
**Confidence:** 3

**Summary:**

The paper introduces GraphPCB, a novel graph-based framework for Printed Circuit Board (PCB) image analysis. It addresses the limitations of standard vision models by converting PCB images into a graph structure (using Voronoi tessellation to define component nodes and spatial edges), effectively capturing structural relationships. The work releases two new public graph-encoded datasets and provides extensive benchmarks using Graph Neural Networks (GNNs), demonstrating that leveraging spatial context significantly improves component classification and aids in downstream tasks like IC segmentation.

**Strengths:**

1. the transformation process: moving from a raw image domain to a topological domain using Voronoi tessellation to model component proximity.
2. Applying Graph Neural Networks (GNNs) to PCB component is a new applicatoin

**Weaknesses:**

1. Lack of novelty: The use of Voronoi tessellation to define spatial proximity and generate a graph from point data (component centroids) is a well-established technique in computational geometry.

2. Feature Extraction is Off-the-Shelf: The method for generating node features involves using a pre-trained ResNet50 CNN, which is a standard, off-the-shelf feature extractor from the computer vision domain. There is no novel feature engineering or architecture proposed for generating the visual embeddings tailored to the PCB task.

3. The core classification is performed using widely accepted, foundational Graph Neural Network architectures (GCN, GAT, GraphSAGE). While providing a strong benchmark, the paper does not propose a novel GNN architecture specifically optimized for the challenges of PCB graphs

4. The contribution can be viewed primarily as a system integration effort—successfully combining component localization, standard graph generation, off-the-shelf feature extraction, and GNN classification—rather than  in algorithmic novelty.

**Questions:**

q1: The entire framework relies on precise component localization. How sensitive is the final GNN classification accuracy to noise or slight inaccuracies in the component centroid locations, which directly drive the Voronoi tessellation?

q2: Can you provide a detailed discussion on Table 3, does the column means accuracy? why DT Diode columns vary a lot across different method? if it means accuracy, the Recall/POD for critical minority classes like Diode and DTs are consistently low across all GNN models, what are the primary sources of misclassification for these components?

---

### Official Review · Reviewer_zRZz · 2025-11-01

**Soundness:** 2
**Presentation:** 2
**Contribution:** 2
**Rating:** 4
**Confidence:** 3

**Summary:**

This paper introduces a novel framework for converting Printed Circuit Board images into graph-structured data and presents two new datasets, Graph-F and Graph-W, for node classification using Graph Neural Networks. The authors employ Voronoi tessellation to map component regions to nodes and spatial adjacencies to edges, preserving the layout information of PCB components. The paper provides a thorough analysis of graph structural properties (e.g., heterophily, label imbalance) and evaluates a variety of GNN models, demonstrating the potential of graph-based methods in PCB image analysis.

**Strengths:**

(1) First work to model PCB images as graphs and release public datasets, bridging a gap in GNN applications for hardware assurance.

(2) Employs multiple metrics (F1, Subset F1, POD) for a holistic performance assessment.

**Weaknesses:**

(1) Although the authors claim to have released two datasets, these datasets are not originally collected by them but are curated from two publicly available PCB image datasets. Moreover, the scale of the datasets can only be considered moderate. Therefore, I believe this aspect does not meet the standard for a paper to be recognized as a benchmark publication.

(2) The authors need to report the results obtained by training state-of-the-art detection and recognition frameworks on the original image datasets. While the use of a GNN architecture is somewhat justified, there is a lack of comparative experiments to validate its effectiveness. Furthermore, I expect the authors to conduct post-training based on advanced MLLM models (e.g., Qwen2.5-VL or Qwen3-VL) to evaluate their performance, which I consider a necessary comparison.

(3) The GNN model employed in the experimental section is outdated and lacks persuasiveness. The authors should select recently developed advanced GNN models for experimentation. Additionally, as a benchmark dataset paper, the number of compared methods is insufficient, making it difficult to gain an intuitive understanding of the dataset’s quality and complexity.

(4) The use of Voronoi tessellation for graph construction lacks in-depth analysis. The rationale behind the success of this method requires further explanation.

**Questions:**

See Weaknesses

---

### Official Review · Reviewer_XDNN · 2025-11-02

**Soundness:** 3
**Presentation:** 3
**Contribution:** 2
**Rating:** 4
**Confidence:** 2

**Summary:**

This paper creates a dataset for benchmarking Graph Neural Networks by automatically converting printed circuit board (PCB) images into graphs and tasking networks with classifying nodes of the graph to different components, with component locations and labels coming from a previous PCB image classification datasets. The authors analyze structural properties of the created graphs, and show that the dataset contains desirable properties for a graph ML dataset such as diversity, heterophily, and high label informativeness. The authors evaluate GNN baselines on the node classification task, and find GraphSage generally performs best. Last, the authors apply GraphSage on the created dataset as part of an IC segmentation task.

**Strengths:**

Strengths:
- Component classification of PCB's is an interesting domain, with interesting challenges for image recognition models.
- The idea of classifying via an intermediate graph structure is an interesting idea to pursue and evaluate
- Likewise, the idea of creating a dataset for evaluating GNN's from PCB's is great
- The approach for creating a graph dataset is reasonable
- The dataset structural analysis is very thorough
- The evaluation of baselines is solid

Note: this paper is outside my area of expertise, so it is hard for me to substantively evaluate the paper especially in relation to related work and standards for the area (GNN benchmarks)

**Weaknesses:**

Weaknesses:
- Several details behind the dataset creation are unclear (see questions section)
- The dataset created may lack significance or impact. As someone outside of the field of GNN's, it's hard for me to evaluate the significance of the new dataset and how it stacks up to related work.
- The purpose of Section 4.4 is unclear (see questions)

**Questions:**

Q1. how many data points are in each dataset?

Q2. what do the FPIC and WACV datasets contain? labels of components, and also bounding boxes of the components? or just coordinate locations of the components? — how precisely is this information used when doing the Voronoi partitioning?

Q3. Is it common for GNN datasets to be made "synthetically" this way?

Q4. Is the main benefit of this dataset providing a benchmark for GNN's, or could there be any benefit to use this dataset to train for PCB?
My impression is that it's only the former, since component location information is used to create the graph structure.

Q5. Help me understand Section 4.4 better. Can you explain why SSR-Sage is not a rigorous IC segmentation model? Here is my impression of what is going on:
- What dataset is SSR-Sage being evaluated on? Is it the same dataset(s) that the graph dataset is created with?
- Why can't SAGE perform object detection? Isn't this the same as node classification?
- My impression is that to perform classification, the SAGE model first gets the graph conversion of the image, and then classifies nodes.
- Maybe this is why it's not a rigorous model: because converting to graph from uses the node locations and labels, so you're already "baking in" information about the ground truth into the model's predictions.
- If so, then this whole approach seems useless, so I'm not sure why it's included. I realize the authors already use cautious language in section 4.4 ("it is not a rigorous model" but unclear why it's being considered as a task then. Help me understand better

---

### Note · Authors · 2025-11-15

I have read and agree with the venue's withdrawal policy on behalf of myself and my co-authors.